

# To call a cloud 'cirrus': sound symbolism in names for categories or items

VanjaKović[1], Jelena Sučević[2] and Suzy J. Styles[3]

[1] Department of Psychology, University of Belgrade, Belgrade, Serbia
[2] Department of Experimental Psychology, University of Oxford, Oxford, United Kingdom
[3] Division of Psychology, Nanyang Technological University, Singapore, Singapore

## ABSTRACT

The aim of the present paper is to experimentally test whether sound symbolism has selective effects on labels with different ranges-of-reference within a simple noun-hierarchy. In two experiments, adult participants learned the make up of two categories of unfamiliar objects ('alien life forms'), and were passively exposed to either category-labels or item-labels, in a learning-by-guessing categorization task. Following category training, participants were tested on their visual discrimination of object pairs. For different groups of participants, the labels were either congruent or incongruent with the objects. In Experiment 1, when trained on items with individual labels, participants were worse (made more errors) at detecting visual object mismatches when trained labels were incongruent. In Experiment 2, when participants were trained on items in labelled categories, participants were faster at detecting a match if the trained labels were congruent, and faster at detecting a mismatch if the trained labels were incongruent. This pattern of results suggests that sound symbolism in category labels facilitates later similarity judgments when congruent, and discrimination when incongruent, whereas for item labels incongruence generates error in judgements of visual object differences. These findings reveal that sound symbolic congruence has a different outcome at different levels of labelling within a noun hierarchy. These effects emerged in the absence of the label itself, indicating subtle but pervasive effects on visual object processing.

Corresponding author
Vanja Ković, vanja.kovic@f.bg.ac.rs

## INTRODUCTION

When cloud watchers talk about a 'cirrus' or a 'cumulus', they are naming cloud types which can be distinguished by their visual form. The cirrus (with its voiceless sibilants, and high front vowel), refers to a high-altitude ice cloud, with a typically thin, wispy form. The cumulus (with its sonorant consonants, and rounded vowels), refers to a puffy, low-altitude cloud, prone to rising muffin-tops. Do the sounds encoded in the names of the clouds enhance visual differences between the clouds' shapes? And if so, does an alliance of sounds with visual forms differ when labels are used at different levels of description (e.g., 'cloud'

[1] Although this lexical pair is not a canonical example of linguistic sound symbolism, when presented with a pair of pictures showing canonically wispy cirrus (cirrus uncinus) and canonically puffy cumulus (cumulus congestus) clouds, 83% of 135 Serbian speakers unfamiliar with the formal scientific cloud names guessed which cloud is which. For further details, see the Open Science Framework repository for this article (https://osf.io/2wvug/).

versus 'cirrus')? This nebulous example[1] allows us to ask an important question about the relationship between labelling itself, and a label's 'range of reference,' for items that can be named at different levels within a noun hierarchy.

Since the first systematic studies into sound symbolism in linguistic labelling, research has tended to focus on the mapping of labels onto a small number of individual items (usually a pair), which can be discriminated on some visual dimension. For example, *Sapir (1929)* asked people which of the word-forms 'mil' or 'mal' would be a better label for pairs of items like a small or large table, and *Köhler (1929–1947)* asked whether a curved or an angular line-drawing would be a better match for word forms like 'baluma' and 'takete'. The majority of the literature on sound symbolism (for recent reviews see *Imai & Kita, 2014*; *Lockwood & Dingemanse, 2015*) has followed this trend for contrastive label-stimulus mappings, typically for a small number of items, usually two-dimensional line drawings, usually presented in pairs (prominent examples include *Maurer, Pathman & Mondloch, 2006*; *Ramachandran & Hubbard, 2001*). Some alternative approaches include matching single words with pairs of pictures randomly generated to match certain visual characteristics of known congruence types (*Drijvers, Zaadnoordijk & Dingemanse, 2015*; *Nielsen & Rendall, 2011*; *Nielsen & Rendall, 2013*), or matching word-forms to a variety of natural and artificial objects, with varying agency (*Flumini, Ranzini & Borghi, 2014*). However, as the majority of methods rely on the one-to-one match between a single label and single visual stimulus, it is difficult to interpret what range-of-reference these novel labels might play in naturalistic noun-labelling hierarchy–do they refer to abstract superordinate categories (e.g., animal), basic-level categories (e.g., dog), subordinate-nouns (e.g., Labrador), or even proper nouns for individual exemplars (e.g., Fido).

Research using words from unfamiliar languages provides a ready-made range-of-reference, since participants will likely bootstrap the novel word to its translation equivalent. For example, in a study by *Berlin (1995)*, English-speaking participants were above chance at guessing which label in a pair was a type of fish, and which a type of bird, for stimuli from an unfamiliar language. Studies of this kind demonstrate that sound symbolism can reside in labels for categories of natural objects, not just as one-off names for arbitrary experimental tokens: sound symbolism can enhance word learning for antonym pairs in unfamiliar languages (e.g., 'fast'/'slow') (*Lockwood, Dingemanse & Hagoort, 2016*; *Nygaard, Cook & Namy, 2009*), and facilitate word learning in tasks where adults and infants learn labels for novel action events, and extend them to new exemplars of the action (*Imai et al., 2008*; *Kantartzis, Imai & Kita, 2011*). Sound symbolism is thus not limited to arbitrary lab-based labels for abstract visual items, but is evident in natural-category labels, descriptive phrases, and action descriptions. However, among this body of research, it has remained unclear whether sound symbolism plays different roles at different levels of labelling within a single hierarchy.

There is a growing body of research suggesting that words have a special status in activating conceptual representations (*Lupyan & Thompson-Schill, 2012*). For example, labels activate representations that are more abstract (*Edmiston & Lupyan, 2015*), and facilitate acquisition of novel categories even when the label is redundant to the category (*Lupyan, Rakinson & McClelland, 2007*), whereas this is not the case for other correlated non-verbal cues, such as environmental sounds. The speech-specific nature of labelling

on category formation has been shown in infants as young as 10-months-of-age (*Althaus & Westermann, 2016*), where labels have been shown to induce visual attention to parts of objects containing featural similarities (*Althaus & Mareschal, 2014*). That is to say, even for babies, hearing objects named-alike induces attention to shared object properties, and these attentional biases are thought to underly the process of categorization. Bringing categorization and sound symbolism together, a recent word learning study by Monaghan and colleagues (*Monaghan, Mattock & Walker, 2012*) showed that sound-symbolic labels facilitated learning the mapping between the label and the *category* of items, but not between the label and the *individual* objects.

One series of studies (*Kovic, Plunkett & Westermann, 2010*) has investigated sound symbolism in labels which explicitly refer to whole categories of novel objects where multiple members of the category are available to the participant. In *Kovic, Plunkett & Westermann (2010)*, participants learned to sort two classes of novel, schematic creatures, by allocating them to named categories. Participants began by guessing which creature belonged to which category, and with feedback, gradually refined their answers as they learned to categorize items. The creatures differed according to a number of schematic dimensions (e.g., number of legs, shape of tail), but no single feature was solely responsible for which item belonged to which category. Two groups of participants learned to form categories with labels which were sound-symbolically congruent or incongruent with the typical shape of the creatures' head (e.g., 'dom' or 'shick' for mostly-round or mostly-pointy heads). In two separate behavioural tasks, the pairings of labels to categories did not influence training speed or recall accuracy. However, during later test phases (in which people heard a label, then decided whether a subsequently presented picture matched) the congruence of the shape of the creatures' heads with the newly learned labels influenced the *speed of decisions*. This finding suggested that although the efficiency and success of word learning did not differ between conditions, links between the auditory label and the category members were more efficiently accessed for participants who had learned sound symbolically congruent labels.

The same study also reported electrophysiological evidence of differences between responses of the two groups of participants—during the test phase, ERPs showed characteristics of enhanced visual object processing for people who had learned the congruent category labels. Taken together, these findings suggest that when sound symbolism is present for a recently acquired category label, it enhances our ability to access information relevant to the mapping between form and meaning. In terms of mechanisms, this finding could suggest that a congruent label invokes (primes) salient features of its member items prior to their visual appearance, thus enhancing object recognition. Alternatively, encoding with symbolism may cause the individual items to be represented with richer feature representations, meaning that their individual shapes were recognized more efficiently. This could be achieved by more efficient access to the stored knowledge about each item during the test phase, following sound-symbolic encoding during training.

However, notable questions still remain about the limits of sound symbolism in naturalistic language environments. As argued in a recent review (*Dingemanse et al., 2015*), if all concepts with similar meanings had similar-sounding names, the phonological

space for semantically related items would become extremely crowded, leading to a kind of 'systematicity' in which multiple similar form-meaning relationships generate high confusability. This crowding may necessitate a tug-of-war between systematicity and arbitrariness in natural language systems, with systematicity arising from similar meanings being encoded by similar sounds, and arbitrariness facilitating discriminability between items with similar meanings. It has therefore been suggested that systematicity plays a role in grouping together categories, rather than distinguishing between exemplars. This claim has been tested on meaning relations arising from different grammatical categories (*Monaghan & Christiansen, 2006*; *Monaghan, Christiansen & Fitneva, 2011*). However, it remains to be tested whether similar effects are observed for labels at different levels of noun taxonomy.

The aim of the present paper is to experimentally test whether sound symbolism has a different effect in labels with a different range-of-reference within a simple noun-hierarchy. We conducted a study in which people learned the make up of two categories of unfamiliar objects ('alien life forms'), and were passively exposed to either labels for the categories, or labels for each item, while they learned. Within this framework, for different groups of participants, the labels were either congruent or incongruent with the objects. Since previous research (*Kovic, Plunkett & Westermann, 2010*) has demonstrated that sound symbolism does not alter the speed or the accuracy of category formation, we did not predict the influence of sound symbolism to be present during training, but to emerge in tests conducted after category learning. We therefore asked whether congruence between the label and the visual form during training might cause people to attend to the objects differently, and influence visual object processing, even in the absence of the auditory training stimulus (the label). The test phase was implemented as a silent, visual same/different judgement, to tease apart whether effects are driven by sound symbolism altering the strength of the mapping between the word-form and the visual features of the object, or by sound symbolism altering the representation of the visual objects themselves. To the best of our knowledge, no other studies have attempted to look at differences in visual object processing caused by sound symbolism, in the absence of the sound symbolic stimulus.

We predicted that if sound symbolism is effective in highlighting featural similarities among members of a category, then congruent labels would enhance performance (faster RT and fewer errors) for categories learned with category-labels, but would not influence performance for categories learned with item labels. On the other hand, if sound symbolism enhances object processing on a trial-by-trial basis, via a generalized form of crossmodal congruence between individual sounds and individual object features, we predicted that training with congruent labels would enhance performance (faster RT, fewer errors) for categories learned with all levels of labelling.

## METHOD

### Participants

Eighty-two participants from a local Science Center and second-year psychology undergraduate students at the first author's institution (13 male, 17–19 years old), took part

in the present experiment and received course credit for their participation. All participants reported normal or corrected-to-normal vision and were unaware of the purpose of the present study. This research was approved by one branch of the Serbian Psychological Association, the Ethics Committee from the Department of Psychology, University of Niš (document number 3/16).

## Stimuli

Visual stimuli were abstract pencil drawings of novel complex shapes designed for use in a number of different studies. The forms were inspired by the shape of the historic Vinča figures, products of a Neolithic culture from the Balkan region. These abstract drawings were created to form two categories, differing in their visual properties: one category consisted of curved shapes, in rounded forms, and the other category, angular shapes, in a vertical orientation (see Fig. 1B). Each category contained six members. Given their abstract forms (reminiscent of Picasso), we term these stimuli Vinčasso illustrations.

Given the novelty of stimuli, a rating task was run in order to establish that the two sets of Vinčassos did indeed represent a contrast between soft/round and sharp/angular visual forms, with between category difference, and within-category homogeneity. 135 psychology and education second year undergraduate students (age: 17.5–19.5 years) were asked to judge each item's shape on a 7-point scale ranging from 'round' (at the leftmost side of the scale) to angular (at the right). The online rating task was implemented using the Qualtrics survey platform, where each item was presented in a random order. The results showed that the pencil drawn Vinčassos did indeed fall into two categories (Round: $M = 1.58$, $SD = .10$; Sharp: $M = 5.68$, $SD = 0.43$; $t(5.51) = 22.98$, $p < .001$), with high between category difference, and high within-category homogeneity. The same participants also rated the complexity of the pictures from 'simple' to 'complex'. The results showed no overall differences between the two Vinčasso categories (Round: $M = 4.3$, $SD = .79$; Sharp: $M = 3.94$, $SD = .45$; $t(1, 11) = .95$, $p = .36$), which showed substantial within-category and within-subject variability. The original Vinčasso stimuli at screen-resolution can be found in the Open Science Framework repository for this article (https://osf.io/2wvug/), and further details of the rating study can be found in Supplemental Information 1.

Twelve auditory pseudowords were used as labels for the pictures. The pseduowords have a C-V-C-V-C structure, and were read aloud by the first author, a female native speaker of Serbian, using a neutral intonation. In the current study, we elected to use natural speech in preference to artificially generated speech, as we are interested in the naturalistic processing of realistic whole language, without the potential confounds introduced by participants' awareness of the unnatural characteristics of synthesized or flat-prosody speech. Stimuli were recorded in a single session and filtered to remove hiss and hum using GoldWave. Stimuli were approximately equal in duration ($M = 790$ ms, $SD = 82.3$ ms). Half of the stimuli contained soft phonological structure (e.g., 'volab') and the other half, sharp phonological structure (e.g., 'šičak'; see Fig. 1 for further details). The 'sharpness' and 'softness' of the phonemes were based on findings of several previous studies of Serbian, the language of test, and the twelve test items were previously validated for their 'sharp' and 'soft' associations (Sučević et al., 2015). The audio files used for presentation

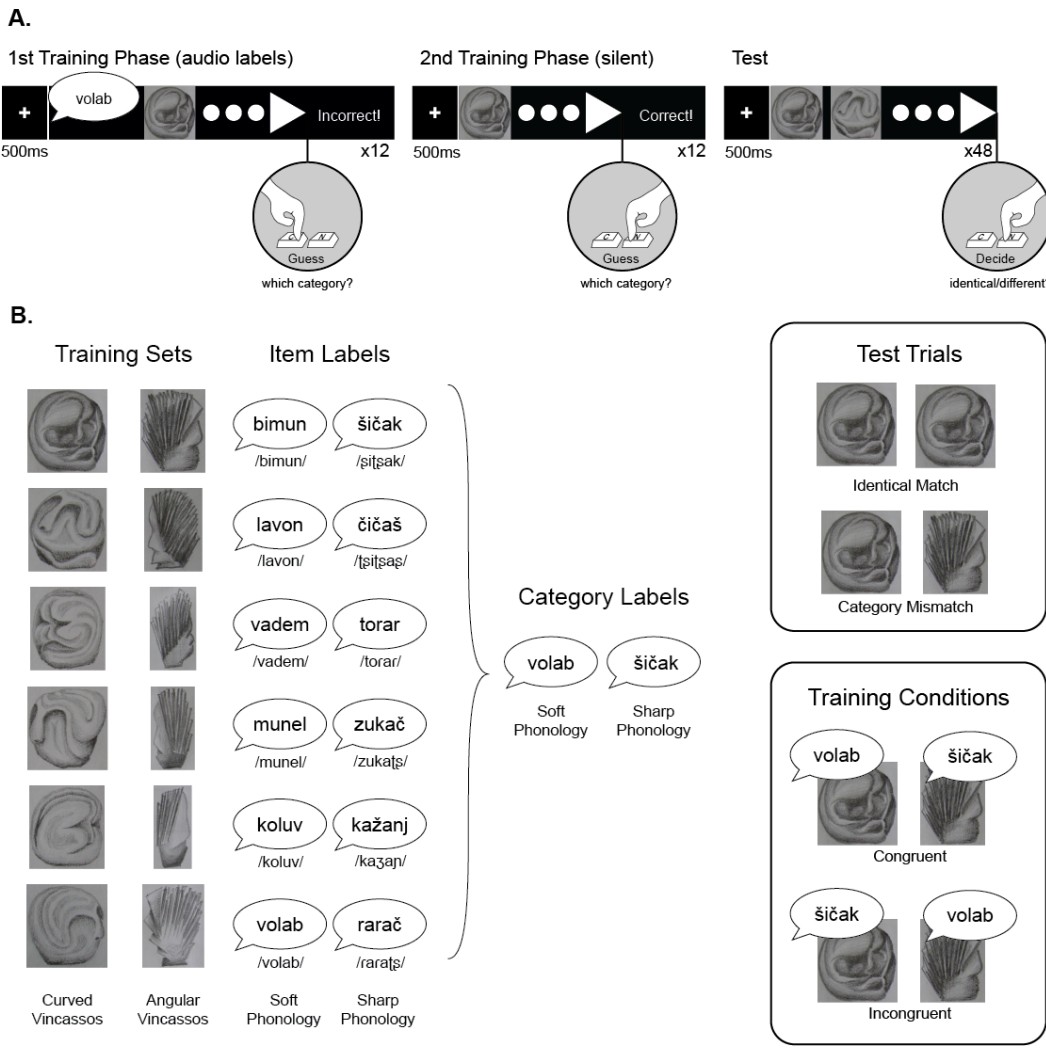

**Figure 1 Experimental design.** (A) The three blocks of the study, showing the structure of trials, and event timing. (B) Stimuli and experimental conditions. Structure of the visual categories, and the twelve auditory labels used in Item Label training condition (names are given in standard Serbian Latin orthography, and glossed in IPA), and the 2 labels used in the Category Label training condition, along with examples of the test trial types, and the training conditions.

in the experiment can be found in the Open Science Framework repository for this article (https://osf.io/2wvug/). As it has been suggested both phonetic and prosodic characteristics combine to generate sound symbolic effects (*Dingemanse et al., 2016*; *Nygaard, Cook & Namy, 2009*), further details about the acoustic envelope of the stimuli can be seen in Supplemental Information 2, where the stimuli for the different shape conditions show substantial overlap in their prosodic contours.

## Experimental design and procedure

Participants were asked to imagine they have just landed on an unexplored planet and they need to learn to differentiate between two species that live on the planet in order to survive.
Participants were seated in front of a computer and were given headphones. Instructions were presented on screen.

The experiment consisted of a training and a test phase. In the training phase, participants were familiarized with the categorical structure of the presented objects. In the first half of the training phase, members of the two categories were presented one-by-one, with an auditory label: each trial began with a fixation cross presented for 500 ms ($\pm$100 ms), followed by the auditory label, then a single Vinčasso, approximately 12 cm in height. While the creature remained onscreen, participants were asked to guess which of the two categories it belonged to, by pushing one of two buttons ('C' and 'N' on a standard keyboard), with key allocation counterbalanced across participants using ABAB sequence. When participants made their selection, feedback was displayed on the screen for 600 ms ('Correct!' or 'Incorrect!') (Fig. 1). All twelve stimuli were presented once in a random order.

In the second stage of training, labels were omitted, so that the creature was presented immediately after the fixation cross (which remained on the screen for 500 ms $\pm$ 100 ms). All twelve stimuli were presented once in a random order.

After the training phase, participants were instructed to identify whether the two creatures presented side-by-side were the same ('identical') or different. In the Test block, a fixation cross was presented for 500 ms ($\pm$100 ms), followed by the presentation of two Vinčassos side-by-side. The Creatures remained on the screen until participants responded. The test pairs depicted the same item (Identical Match), or items drawn from different categories (Category Mismatch). For mismatching pairs, each image was randomly paired with a picture from the other category, and pairs were yoked across participants. 12 trials in each condition were presented in a random order, making a total of 24 trials of these types.[2]

## Training conditions

*Label Type* was manipulated between experiments: in Experiment 1, participants heard each Vinčasso named with a different label (Item Labels), meaning that the auditory labels were not meaningfully correlated with the to-be-learned category structure. In Experiment 2, participants heard only two labels, each of which labeled one of the categories (Category Labels), meaning that the auditory labels were correlated with the to-be-learned categories, but were redundant, as the task could be solved without them.

*Label Congruence* was systematically manipulated between participants: half of the participants heard labels that were congruent with the visual properties of the Vinčassos (soft phonology—rounded shapes; sharp phonology—angular shapes), and half-heard incongruent labels. Participants were randomly assigned to a Congruence condition.

Since the categories were defined by their visual properties in both experiments (rounded, angular), the labels were an irrelevant feature of the training environment. In other words, participants could have successfully solved the category structure in this task without paying any attention to the labels. Here, we ask the question whether labels presented during a first encounter with a visual object alter subsequent visual processing of the object.

## Data handling and analysis

Mean error rates were computed for individuals in each of the four training conditions, and each of the three types of test trial. Reaction times were transformed into inverse RT

[2]We originally included a third condition in the Test Phase, in which participants saw different pictures from the same category. However, around two-thirds of participants misunderstood the task instructions and answered on the basis of whether the picture was from the 'same category' instead of whether the pictures were 'identical'. The large scale of the dropout rendered this condition un-analysable here (further details can be found in Supplemental Information 3 and the Open Science Framework repository for this project: https://osf.io/2wvug/).
(iRT = 1/RT), to approximate normal distribution for each participant. Based on visual inspection of the distribution, extreme outliers (exceeding 100–4,000 ms) were excluded from further analysis. For correct trials each individual's mean inverse RT was computed. Following analysis, reaction times were back-transformed into milliseconds (RT = 1/iRT) for graphing, and for presentation of means.

For analysis of the training phase, mixed ANOVAs investigated the influence of Training Condition (Congruent, Incongruent) on Error Rate and on Reaction Time across the two training Blocks (First, Second). For analysis of the test trials, mixed ANOVAs investigated the influence of Training Condition (Congruent, Incongruent) on Error Rate and Reaction Time across the two match types (Identity Match, Mismatch). Power analyses were conducted using G*Power 3.1 (*Faul et al., 2009*).

### Experiment 1: item labels

Each Vinčasso was presented with a different label during the first training block. Half of the participants heard congruent, and half, incongruent item labels. Labels were yoked to pictures for all participants in a given condition. The categorization task involved two categories of six visually coherent items (see Fig. 1).

## RESULTS AND DISCUSSION

### Training

Figure 2 shows the results from the training and the test phase of Experiment 1, with the means for individual participants shown separately. Means are given in Table 1. Participants solved the categorization task, with a reduction in errors from the first to the second block (Error: $F(1, 40) = 31.55$, $p < .001$, $_p\eta^2 = .44$, Observed Power = 1.00). No further effects were observed for Error Rate or RT. As seen elsewhere (*Kovic, Plunkett & Westermann, 2010*) in this training-by-guessing paradigm, sound symbolism is not observed to influence performance during training.[3]

### Test

Error Rates differed across the different Match conditions (Match Type: $F(1, 40) = 8.08$, $p = .007$, $_p\eta^2 = .17$, Observed Power = .79), with overall lower error rates for the Mismatch condition than for the Match condition. This effect was larger for participants who were trained with item labels that were congruent with the shape of the individual items, and was absent for participants in the incongruent condition (Match Type × Congruence: $F(1, 40) = 6.38$, $p = .016$, $_p\eta^2 = .14$, Observed Power = .69). This finding suggests that the correct detection of large differences between paired images in the mismatch condition was disrupted by having earlier encountered incongruent labels, suggesting sub-optimal processing of the visual properties of the objects. Rejecting a mismatch would typically be easier in a task of this kind, since low-level visual features (e.g., vertical edges, curves) can provide definitive information about a mismatch drawn from different visual categories. Interestingly, for participants who heard incongruent labels during their training, there was no general difference between match and mismatch trials. Effectively, they were worse than would normally be expected at correctly rejecting mismatches. However, as this effect was

[3]One participant with an unusually high error rate at the end of the second block can be observed in the Incongruent training condition. This participant also has unusually slow RT during training. Removal of this participant from this, and all subsequent analyses, has no impact on the pattern or direction of the effects, with the exception of reduced power in most tests–here, the difference is not notable: $F(1, 39) = 39.83$, $p < .001$, $_p\eta^2 = .51$, Observed Power = 1.00.
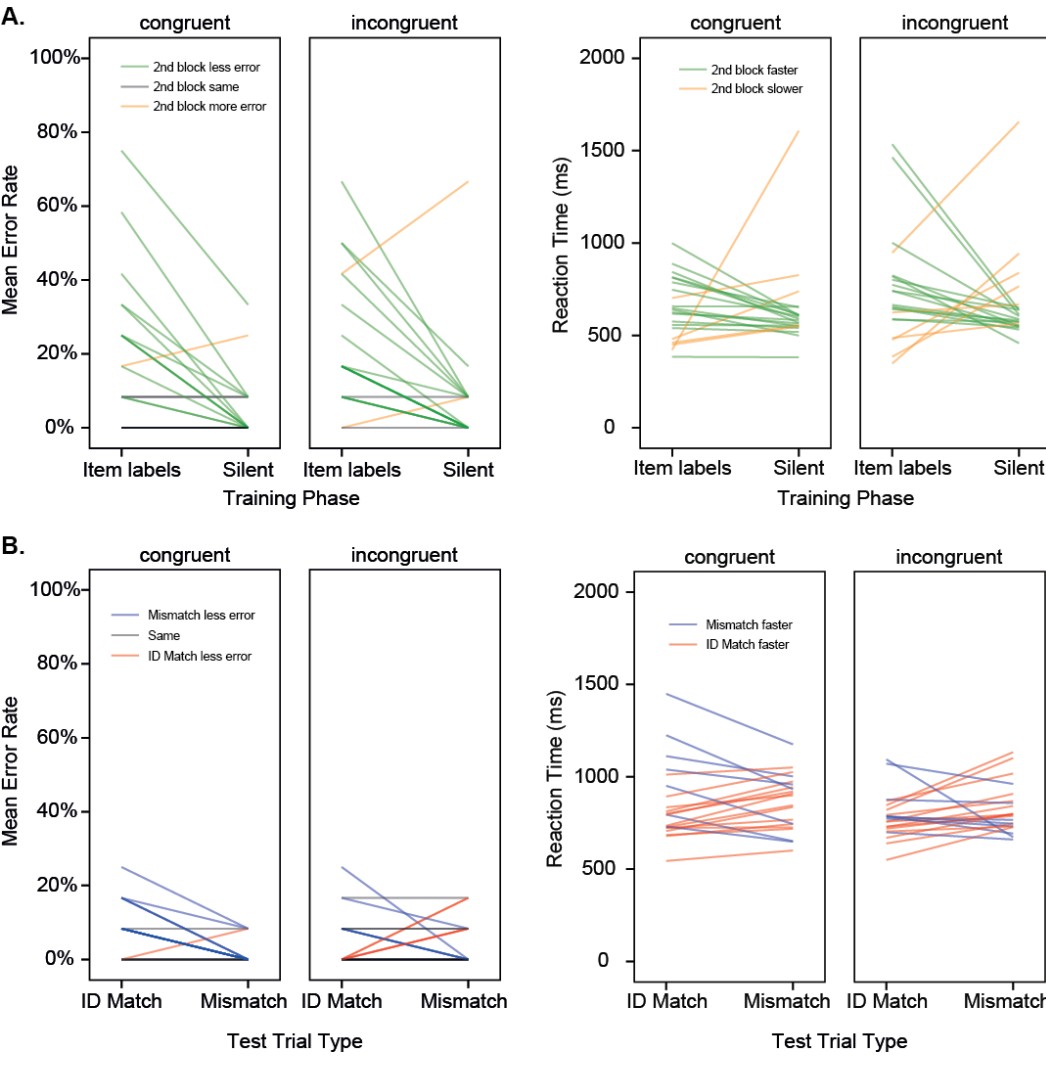

**Figure 2** **Experiment 1.** (A) Results for the two training phases shown for individual participants in different training conditions. Mean error rate, and mean reaction time (back-transformed from iRT) are shown, where GREEN indicates better performance in the second block (more accurate, faster). (C & D). Results for the two test trial types shown for individual participants in different training conditions. (B) Mean error rate, and mean reaction time (back-transformed from iRT) are shown, where BLUE indicates better performance for the Mismatch condition (more accurate, faster). Training with item labels.

[4]In Fig. 2B, this can be observed as the outcome of approximately two thirds of participants showing slightly faster responses for ID match than for mismatch (Congruent: 14, Incongruent, 13), overlapping with the remainder of participants, who showed faster responses for mismatch than for ID match (Congruent: 7, Incongruent: 8), but exhibited a larger reaction time difference between trial types. The degree of individual sensitivity to sound symbolism has previously been shown to correlate with extent of sound symbolic congruence in learning (*Lockwood & Dingemanse, 2015*). Here, since the direction of this split is the same for both training conditions, it does not appear to arise from sound symbolism, but may index a difference in visual object processing.

not predicted, and was somewhat under-powered, it should be treated with caution, and a larger sample size would be required for future investigations seeking to replicate this effect.

Reaction times did not differ significantly for participants in the different training conditions.[4]

## Experiment 2: category labels

Each Vinčasso was presented with one of two auditory labels, each of which labeled a category of six visually coherent Vinčasso creatures during the first training block. Half of the participants heard congruent item labels, and half heard, incongruent item labels. Labels were yoked to categories for all participants in a given condition.
**Table 1  Mean error rates and reaction times (back-transformed from iRT) for Experiment 1.**

| | | Training | | | | Test | | | |
| | | Phase 1 | | Phase 2 | | ID match | | Mismatch | |
| | | M | SD | M | SD | M | SD | M | SD |
|---|---|---|---|---|---|---|---|---|---|
| Error rate | Congruent | .19 | .20 | .05 | .09 | .08 | .07 | .02 | .03 |
| | Incongruent | .22 | .18 | .07 | .15 | .05 | .07 | .05 | .06 |
| RT (ms) | Congruent | 650 | 167 | 637 | 239 | 856 | 212 | 861 | 151 |
| | Incongruent | 750 | 299 | 671 | 251 | 784 | 125 | 825 | 132 |

*Training*

Figure 3 shows the results from the training and the test phase of Experiment 2, with the means for individual participants shown separately. Means are given in Table 2. As in Experiment 1, participants solved the categorization task, with a reduction in errors from the first to the second block (Training Phase: $F(1, 38) = 9.49$, $p = .004$, $_p\eta^2 = .20$, Observed Power $= .85$). However, unlike in Experiment 1, participants exposed to labels correlated with the category structure were, on the whole, *slower* to make their category judgments in the second training block, where category labels were not presented ($F(1, 38) = 7.70$, $p = .009$, $_p\eta^2 = .17$, Observed Power $= .72$). This effect was not predicted, but suggests that in the first block, participants may have been relying on the diagnostic properties of the audio label to make their category guesses, and subsequently found judgments harder in the silent second block when this label was absent. As elsewhere, no influence of sound symbolism was observed on error rates or reaction times during training. As this effect is somewhat under-powered, they should be treated with caution, and a larger sample size would be required for future investigations seeking to replicate this effect.

*Test*

As in Experiment 1, participants made more errors accepting an identical pair than they did rejecting a mismatch (Match Type: $F(1, 38) = 14.00$, $p = .001$, $_p\eta^2 = .27$, Observed Power $= .95$). Unlike in Experiment 1, Reaction Time was influenced by a combination of Match Type and Congruence (Match $\times$ Congruence: $F(1, 38) = 21.89$, $p < .001$, $_p\eta^2 = .37$, Observed Power $= .99$), such that the majority (17 of the 21) of participants trained with congruent labels were faster to accept matches than to reject mismatches, while the majority (17 of the 19) of participants trained with incongruent labels were faster to reject mismatches than to accept matches. This finding suggests that passive exposure to labels during training generates differences in object processing, which are in line with the idea that congruent labels facilitate recognition of salient feature similarities, while incongruent labels highlight featural differences. There was also a general difference between the RTs of people in the different training conditions ($F(1, 37) = 4.65$, $p = .035$, $_p\eta^2 = .11$, Observed Power $= .56$), with people in the incongruent condition slightly faster overall during testing. The low power of this group difference suggests it may be artefactual, and a larger sample size would be required for replication.

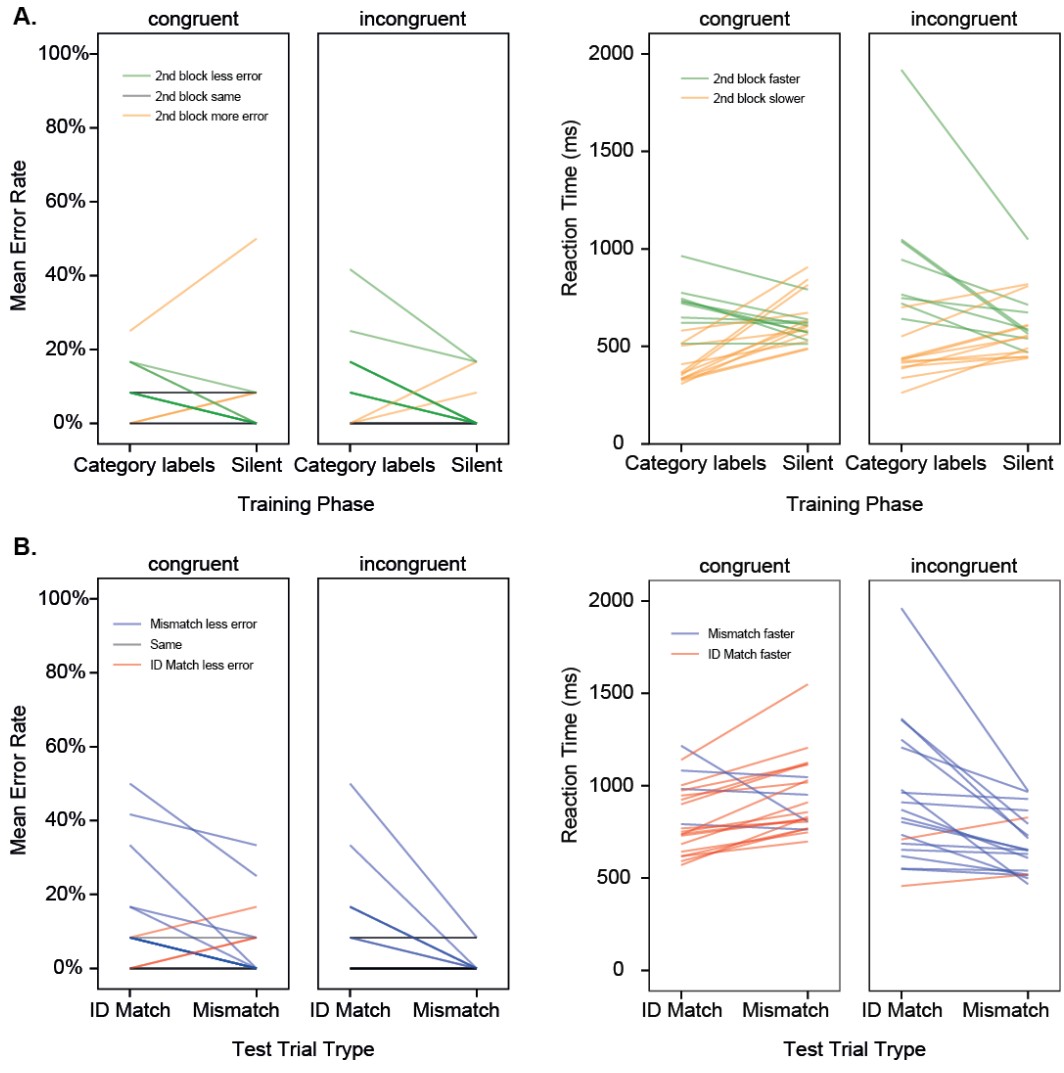

**Figure 3** **Experiment 2.** (A) Results for the two training phases shown for individual participants in different training conditions. Mean error rate, and Mean reaction time (back-transformed from iRT) are shown, where GREEN indicates better performance in the second block (more accurate, faster). (B) Results for the two test trial types shown for individual participants in different training conditions. Mean error rate, and mean reaction time (back-transformed from iRT) are shown, where BLUE indicates better performance for the Mismatch condition (more accurate, faster). Training with category labels.

## DISCUSSION

The key finding of the present research is that the sound symbolism in labels with different ranges of reference (category-label vs item-label) has different outcomes on object processing. Passive exposure to category or item labels during category learning influences later abilities to detect similarities and differences between pairs of visual stimuli: as predicted, category labels influenced the speed of recognizing identical pairs if trained with congruent labels, and the speed of rejecting visually dissimilar pairs, when trained with incongruent labels. In addition, labels disrupted the ability to correctly reject a gross visual difference between two images if the items were trained with incongruent

**Table 2** Mean error rates and reaction times (back-transformed from iRT) for Experiment 2.

| | | Training | | | | Test | | | |
| | | Training 1 | | Training 2 | | ID Match | | Mismatch | |
| | | M | SD | M | SD | M | SD | M | SD |
| Error rates | Congruent | .08 | .07 | .05 | .11 | .11 | .14 | .05 | .09 |
| | Incongruent | .10 | .11 | .03 | .06 | .09 | .13 | .02 | .03 |
| RT (ms) | Congruent | 534 | 193 | 631 | 117 | 829 | 190 | 940 | 204 |
| | Incongruent | 663 | 385 | 601 | 156 | 918 | 371 | 687 | 165 |

labels. These effects are best understood as arising from differences in visual object encoding grounded in the sound symbolism encountered during the initial training stage. Sound symbolism therefore modulates visual object attention differently when labels align with task-relevant category structure.

Until now, the majority of previous studies into sound symbolism have focused on the learnability, guessability or recall of the mapping between a label and its referent. For example, Ković and colleagues *(2010)* demonstrated sound symbolic effects in a test phase where participants heard an auditory label, then saw a matching or mismatching picture; Monaghan and colleagues *(2011)* participants heard an auditory label and guessed from an array of 12 pictures; and Imai and colleagues *(2008)* tested participants on extension of action descriptions to novel agents in a two-alternative forced choice task. In all of these studies, the range of reference of the label was unclear, as there was no experimental control of how labels aligned with other information about category structure.

By contrast, in the current study, we wanted to know whether sound symbolic relations encountered during training would change the way that visual objects were encoded, leading to downstream differences in visual object processing, even in the absence of the label. We found that sound symbolism between a label and a category of objects generated predicted differences in visual processing, but that sound symbolism between individual items and their labels did not generate the enhancements that might typically be predicted, in this categorization paradigm. That is to say, sound symbolism affects the formation of category attributes, not item attributes, and those attributes influence later visual object processing. To the best of our knowledge, this is the first study to show that sound symbolism influences visual object processing outside of the mapping between label and referent.

Participants in both experiments were clearly able to infer the category structure from the trials presented in the first training block, as their accuracy improved in the second (silent) training block, where errors fell to below 10% for participants in all training conditions. In previous research (*Kovic, Plunkett & Westermann, 2010*), sound symbolism did not influence the trajectory of category learning in a learning-by-guessing categorization task, with task irrelevant category labels. In the present research, we replicated and extended this observation, with no difference in category learning for congruent versus incongruent labels, when trained with *either* item or category labels (Experiment 1 and 2, respectively).

Even though the labels presented during the first training block were irrelevant to the task presented in the subsequent test blocks, sound symbolism had pervasive influences

on performance in the silent, visual discrimination task: firstly when trained on categories where each item had different label, participants were worse at rejecting differences than would be typically expected in a task of this kind if they have been trained with congruent labels. This suggests that the normal pattern of object processing may have been disrupted by exposure to incongruent labels. This effect was not predicted, and was somewhat underpowered, so should be treated with some degree of caution. However, if it replicates in future studies, it provides an interesting counterpoint to the perspective that arbitrariness enhances object individuation—it may be the case that *incongruent sound-symbolism* is qualitatively different to other kinds of artbitrariness.

By contrast, when trained on items in labelled categories, participants were better at detecting a match when the labels were congruent, and better at detecting a mismatch when the labels were incongruent (as shown by their RTs). Hence, sound symbolism in category labels enhances similarity judgments when congruent, and enhances object discrimination when incongruent. These results are consistent with the idea that congruent sound symbolism may enhance the processing of shared object features, in a way that enhances similarity judgments, via efficient visual retrieval processes. We also saw incongruent category labels enhance the discrimination of grossly mismatching picture pairs, which may have been due to enhanced recognition of item differences. We therefore see discrete effects of congruent and incongruent sound symbolism at different levels of labelling.

The preceding eighty-odd years of research into linguistic sound symbolism has confirmed that certain sounds in language simply 'go better' with certain perceptual experiences, either via general cross-modal correspondences (*Spence, 2011*), or communication oriented mechanisms of iconicity (*Perniss, Thompson & Vigliocco, 2010*). These effects are known to be automatic (*Parise & Spence, 2012*), unconscious (*Hung, Styles & Hsieh, 2017*), and to enhance memory for individual word meanings under certain conditions (*Imai, Haryu & Okada, 2005*; *Kantartzis, Imai & Kita, 2011*; *Lockwood, Dingemanse & Hagoort, 2016*). The finding of differential effects at different level of labelling draws an interesting parallel with existing research into category learning in the absence of sound symbolism.

Previous experimental and modelling work on massed word learning by *Monaghan, Christiansen & Fitneva (2011)*, has shown that both human learners and neural networks will learn to map word-forms to meaning classes faster if the mappings are systematically linked to the meanings, while learning to individuate specific form-meaning pairs is more effective if the mappings are arbitrary. In their studies, the systematicity arises from sound-alike mean-alike relationships analogous to morphology in linguistic systems like grammatical gender (e.g., prefix X always occurs with word type A). Hence, their kind of systematicity is effective at functionally grouping items which share a similar form, while arbitrariness is better at distinguishing the meanings of individual words. By contrast, linguistic sound symbolism (arising as does from iconic matches between the senses), generally functions at the level of individual pairings between sounds and meanings, rather than at the level of grammatical categories or word classes. Hence, where symbolic congruence gives rise to sound-alike mean-alike relations (due to shared sound-meaning matching processes), it may be responsible for a type of systematicity, as the phonological space available for words with similar meanings becomes crowded by an overabundance of

similar-sounding words. *Gasser (2004)* demonstrated that it is possible to efficiently encode a relatively small number of sound symbolic words. The modelling and experimental work of *Monaghan et al. (2014)* has shown a higher proportion of sound-alike mean-alike relations in the early lexicon than the adult lexicon, suggesting that sound symbolism is most useful when the lexicon is still small. This modelling perspective sits well with findings that sound symbolism can facilitate early word learning (*Imai et al., 2008*), and may act as a 'Bootstrap' for the acquisition of language (*Imai & Kita, 2014*). However, as vocabulary increases, confusability between similar sounding items also increases, as the phonological space becomes crowded, therefore necessitating the emergence of more-arbitrary linguistic structures.

In the current study, our findings therefore support a dominant theoretical perspective in the field of category labelling: that sound symbolism (like other forms of systematicity) does indeed have its greatest impact at the level of Category labels, rather than Item labels (at least in this task, where category structure is task-relevant), but we also show that the effects of sound symbolism go beyond the 'priming' effects which may be generated by a label, or the strength of the link between the label and referent, as they also influence downstream visual object processing, in the absence of the label.

It should be noted that the two testing conditions were not of the same kind: in the Match condition, the decision can be made purely on visual grounds, and the category of the object cannot form part of the decision process. However, in the Mismatch condition, either the learned category difference or the visible item differences could be cues to correctly identifying the mismatch. This discrepancy arises because of an unexpected ambiguity in the wording of out test question, which rendered a third experimental condition un-analysable (for further details, see Supplemental Information 3). This means at this stage we are not in a position to diagnose whether the effects arise out of purely visual processing, or a combination of visual and category cues. Further studies will be needed to clarify this point, along with the remaining question of whether non-sound-symbolic but otherwise systematic labels generate similar effects.

## ACKNOWLEDGEMENTS

Thanks go to Ivana Todorović, who was commissioned to create the original Vinčasso drawings, according to the design parameters and to Andela Šoškić for assisting in the collection of stimulus rating and cloud guessing data. The Cloud Appreciation Society deserves thanks for the final author's sub-ordinate category knowledge of clouds. Preliminary results from some conditions of this project were presented in the poster 'Ković, V, Sučević, J & Styles, SJ "What's in a name?" Category label vs identity label in learning novel categories' at the Dubrovnik Conference on Cognitive Science: Language and Conceptual Development, Dubrovnik, Croatia, May 2014.

### Funding

This research was supported by the Ministry of Education and Science grant No. 179033, Faculty of Philosophy, University of Belgrade to Prof. VanjaKović, and Nanyang Assistant Professorship Grant to Prof. Suzy Styles ('The Shape of Sounds in Singapore', Nanyang Technological University, Singapore). The funders had no role in study design, data collection and analysis, decision to publish, or preparation of the manuscript.

### Grant Disclosures

The following grant information was disclosed by the authors:
Ministry of Education and Science: 179033.
Faculty of Philosophy, University of Belgrade.
Nanyang Assistant Professorship Grant ('The Shape of Sounds in Singapore', Nanyang Technological University, Singapore).

### Competing Interests

The authors declare there are no competing interests.

### Author Contributions

- VanjaKović conceived and designed the experiments, performed the experiments, analyzed the data, contributed reagents/materials/analysis tools, wrote the paper, reviewed drafts of the paper.
- Jelena Sučević conceived and designed the experiments, performed the experiments, analyzed the data, wrote the paper, reviewed drafts of the paper.
- Suzy J. Styles conceived and designed the experiments, analyzed the data, wrote the paper, prepared figures and/or tables, reviewed drafts of the paper.

### Ethics

The following information was supplied relating to ethical approvals (i.e., approving body and any reference numbers):

The Serbian Psychological Association from the University of Niš, Department of Psychology granted Ethical approval to carry out the study within its facilities (document number 3/16).

### Data Availability

Kovic, V., & Styles, S. J. (2017, June 8). Sound Symbolism for category names and item names with Vincasso stimuli. Retrieved from osf.io/2wvug.

### Supplemental Information

Supplemental information for this article can be found online at http://dx.doi.org/10.7717/peerj.3466#supplemental-information.

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
