# Peer review of "To call a cloud ‘cirrus’: sound symbolism in names for categories or items"

_PeerJ, doi:10.7717/peerj.3466_

## Round 0.1 · original submission · Major Revisions

I have now received two reviews from experts in the field. I thank the reviewers for their work. Both reviewers think that your approach is interesting and innovative, but both raise a number of concerns that need to be addressed for a successful revision. Based on my own independent reading of your work, I agree with their evaluation. In the rest of my action letter let me focus on the issues that I consider crucial for a revision; for the rest please refer to the reviews, that are thoughtful and quite clear.

Clearer conceptual framing. The conceptual framing underlying this work needs to be made more explicit and the theoretical background needs to be strengthened. This can be achieved by improving grounding of your work in current literature (see comments of both reviewers), by using a better defined terminology, and by better highlighting the novelty of your approach, in which you focus on sound symbolism’s impact on categorization. Crucially, your predictions should be clearer and more compelling, and offer a way to interpret the results you find.

Stimuli selection. The rationale underlying the selection of your stimuli should be better spelled out (see comments of reviewer 2); furthermore I agree with reviewer 1 that you will need to rate your stimuli.

Methodological details. You might want to use more than one figure to help the readers, and to offer the reader more details as to means and data distribution (see comments of both reviewers).

·

Basic reporting

The paper raises two basic questions: (1) Do differences in the form of labels influence the perception of differences between what the labels refer to? (2) And does this influence differ based on the level of categorization (e.g. an item vs. its superordinate category)? It studies these questions experimentally with a set of visual stimuli and pseudowords applied to items and categories. I am sympathetic to this line of work as I agree with the authors that too much of the pseudoword sound-symbolism literature has focused on reference rather than categorisation. However, the ms in its present form has a number of flaws that prevent me from recommending publication.

One of the most important issues is embedding in the literature. There is a set of experimental studies by Gary Lupyan and colleagues which is highly relevant and overlapping in many respects: using nonwords, contrasting labels for categories and items, and even using ‘alien life forms’ as experimental stimuli (Lupyan, Rakison, and McClelland 2007; Lupyan 2008; Lupyan and Casasanto 2014; Edmiston and Lupyan 2015). In a related study, Lupyan and Thompson-Schill (2012) examine the distinction between the sound of a dog bark (which evokes a single member of a category) versus the word “dog” (which evokes a more categorical representation). Another highly relevant study by Monaghan and colleagues tests novel word learning at category level and at item level (Monaghan, Mattock, and Walker 2012). Together, these studies offer conceptual foundations, methodological approaches and empirical results that are directly relevant to the project reported in the present ms. So while the present ms offers converging evidence for some the findings from this rich line of work, the claims of priority and uniqueness (e.g. lines 67-68, 318-319) seem wholly unwarranted, and the reporting of the work should be thoroughly reexamined in light of this prior work to situate and contextualise the scholarly contributions as appropriate.

PeerJ recommends the public deposition of high quality versions of figures, visual and auditory stimuli and raw data. None of these were provided in the materials supplied to reviewers, and the ms does not discuss whether the data is open. This should be improved. Consider sharing materials through Open Science Framework (osf.io) or through the PeerJ system.

The main and only Figure is very complex and combines information that belongs in different sections of the paper. I would recommend breaking it up accordingly, which also makes it possible to have more informative captions for each panel.
Panel C present bar plots, which reduce complex data to means and error bars and thereby obscure distributional information that could be crucial. I would recommend making the visuals more informative by plotting the data as scatterplots, density plots or boxplots — see here for more info: https://jimgrange.wordpress.com/2016/06/15/solution-to-barbarplots-in-r/. Plotting distributions rather than just summaries may also reveal individual differences, which have recently been shown to correlate with sound-symbolic sensitivity (Lockwood, Hagoort, and Dingemanse 2016).

As a minor comment, there is probably a bit too much poetic license in the introductory metaphor (lines 31-34), which is unfortunate because it recurs in the title. There is only one high vowel /ɪ/ in cirrus [ˈsɪrəs], so the word is not “full of them”. There is only one rounded vowel /u/ in cumulus [ˈkjuːmjʊləs], the rest is unrounded. The number of sonorant consonants is the same in both (/r, m/), so that doesn’t distinguish cumulus from cirrus. It is anyway unclear whether this slight contrast would elicit the same kind of result the authors report; and since the focus of the ms is on pseudowords and novel referents, it seems unnecessarily confusing to elevate that somewhat shaky metaphor to the the prime example of the contrast targeted experimentally.

Experimental design

The study uses a set of newly created “abstract pencil drawings of complex shapes … created to form two categories” (as noted above, the drawings are not made available separately, which should be rectified). The two categories are described as [i] “curved shapes, in rounded forms” and [ii] “angular shapes, in vertical orientation”. There are several potential issues with the stimuli which are not discussed and potentially have important consequences. It seems a separate rating experiment would be required to establish that the two sets of visual stimuli indeed represent soft/sharp or round/angular concepts, and are different from each other on just these parameters while also showing sufficient within-category homogeneity. There are also various differences in the sets of drawings that introduce confounding cues: the stimulus sets differ in size and/or orientation, as well as potentially in visual complexity (again, an independent rating study should be used to establish whether this is the case). With so many possible differences between the sets of stimuli, it is unclear what the basis is for categorizing the presumed sound symbolic mappings as being about curved or angular.

Similar problems hold for the auditory stimuli. Labels are claimed to be congruent or incongruent with the visual properties of the pictures, but no independent rating study provides evidence that they are indeed congruent, and even if so, whether there are differences in congruence across label/picture pairings (the training and test data may provide some indication here, but the results are not analysed by item). Labels are nonwords recorded by a native speaker of Serbian. Was the speaker aware of the experimental manipulation? If so, there is a real chance of introducing experimenter bias. Even if not, there are many other things that should be controlled for. The items are designed to be contrastive in “soft phonological structure” versus “sharp phonological structure”, but it is not clear that this is the only relevant difference between them. Length is controlled for, but no controls or separate ratings or measures are reported for prosody and a host of other phonetic features that may or may not provide conflicting or confounding cues. Labels are presumably given in Serbian orthography; it is not clear how this relates to phonetics, and so for reproducibility, IPA renditions should be given at the very least (much better would be to share the auditory stimuli themselves). See Nygaard et al. (2009) and Dingemanse et al. (2016) for why controlling for prosody is just as important as controlling for segments in experimental studies of sound symbolism. See Monaghan (2012) for an example of the things to be controlled for in auditory stimuli when not using speech synthesis.

Rating and measuring experimental stimuli prior to experimental work is standard fare in psycholinguistics, especially when novel stimuli are introduced. The authors do themselves a disservice by not doing this, as this makes it unclear what exactly the results can be ascribed to. As it is, the possibility cannot excluded that any of a number of possible confounds play a role in obtaining the results. Incidentally, many of these challenges could be avoided by using a well-known and well-controlled set of existing stimuli such as the YUFO stimulus set (Gauthier et al. 2003). The benefit of using preexisting stimuli is that it is clearer how experimental findings relate to and build on prior work. There can be a benefit to using novel stimuli too, especially if they allow one to address a new research question and if they are made available to the scholarly community — but neither of these seems to be the case for the present ms.

There are also some questions about the design. One thing that is not clearly explained or motivated is why an identical match always literally the same item (as depicted in Panel B, Fig 1), rather than an item from the same category. This decision is of a very different kind than the categorical decisions participants been set up for in the training task. Footnote 1 reveals there was another condition where “same” stood for “same category”, but that responses to this were hard to analyse. It is unclear how much data was excluded as a result; this should be reported. The result is that the two testing conditions are not of the same kind: in the Match condition, category doesn’t matter at all as the decision can be made solely on visual grounds; in the Mismatch condition, both the learned category difference and the visible item difference can be cues.

Validity of the findings

Due to the lack of embedding in prior work and the potential confounds in the visual and auditory stimuli, it is not really clear to this reviewer how the findings are to be interpreted. What makes interpreting the findings harder is that few clear predictions are made at the outset; the most specific we get is that if sound symbolism serves to highlight similarities between category members “an advantage … will be observed for category-label learning but not item-label learning”, whereas “if the effects of sound symbolism are pervasive wherever they arise, we predict that [its] influence will be present in all labeling conditions”. For the latter case, the conclusion is merely a redescription of the antecedent conditional (“if sound symbolism is pervasive, its influence is everywhere”). For the former, it is not specified at the outset how the advantage will be measured. Any measurement will result in at least some differences; the direction in which these differences go matter a lot. For instance, one might predict that learning labels that capture similarities between within-category items (‘congruent’) might make it easier to distinguish between items from different categories, and that this would be reflected in lower RTs and lower error rates. But no such predictions are made.

Only later we learn that error rates and reaction times show different patterns, and now it is not clear to what extent their interpretation is more than a post hoc rationalisation. For instance, the highest RTs of all (assuming that the barplots don’t hide a skewed distribution) are in Exp 2, Mismatch-Congruent condition: so apparently, deciding two items are different is harder when you’ve previously learned category labels that fit will (are congruent). It is not clear how this follows from any of the predictions, and therefore the ms is reduced to claiming this is evidence that exposure to labels “generates differences in object processing”. Without proper grounding in the broader literature, this doesn’t really bring us closer to a mechanistic account of sound-symbolic effects.

Another challenge in interpreting the findings is that the discussion in the ms is equivocal on key concepts and interpretations. For instance, the term ‘iconicity’ (usually regarded a superordinate term of sound symbolism) is first introduced on p 12, almost as an afterthought, which results in the following hard to interpret passage: “Systematic sound symbolism improves detection of visual similarities for category labels, while iconicity enhances object discrimination for both item labels (less errors), and category labels (faster responses).” If the authors mean to contrast “systematic sound symbolism” with “iconicity”, it is not clear what either of these means and how they are different.

Part of the problem, one suspects, is the lack of a clear distinction between “sound symbolism” and “systematicity”. Some of the Monaghan papers cited in the discussion are more about systematicity than about iconicity; the distinction is hashed out in the cited TiCS review by Dingemanse et al., but its implications (in particular, that systematicity involves statistical regularities at the level of word categories while iconicity involves perceptual analogies at the level of word meanings) are not fully worked out.

Also hard to understand is the use of the term “sound symbolism” as if it were an attribute of labels in isolation. So “sound symbolism in category labels” is described as being “congruent” or “incongruent”. But surely sound symbolism does not inhere in labels itself but only in the relation between labels and their referents (be it category or item) — so it is somewhat of an oxymoron to speak of a category label that “has sound symbolism” yet is “incongruent”. The reader has to do a considerable amount of mental gymnastics to follow the argument, and whereas normally one can turn to the discussion for the clearest statement of the predictions and findings, here the discussion rather adds to the confusion by introducing a further set of terms and distinctions which are not entirely in line with those in earlier parts of the study.

Additional comments

In closing, let me reiterate that I am sympathetic to the research program and I do think there is interpretable data here. What is missing is a clear conceptual framework, rooted in prior work, that will help to interpret the results that can make sense of them in light of related findings. If this is supplied, and if measurements or rating studies on the stimuli can show that the effects stand when confounds are controlled for, this paper may make a contribution to the literature on category effects in sound symbolism. As it is, however, the paper is not there yet. I offer these notes in the hope that the authors will benefit from them in revising and rethinking the work.

Here are the references cited in the above fields:

Dingemanse, Mark, Will Schuerman, Eva Reinisch, Sylvia Tufvesson, and Holger Mitterer. 2016. “What sound symbolism can and cannot do: testing the iconicity of ideophones from five languages.” Language 92 (2): e117–e133. doi:10.1353/lan.2016.0034.

Edmiston, Pierce, and Gary Lupyan. 2015. “What makes words special? Words as unmotivated cues.” Cognition 143: 93–100. doi:10.1016/j.cognition.2015.06.008.

Gauthier, Isabel, Thomas W. James, Kim M. Curby, and Michael J. Tarr. 2003. “The Influence of Conceptual Knowledge on Visual Discrimination.” Cognitive Neuropsychology 20 (3–6): 507–523. doi:10.1080/02643290244000275.

Lockwood, Gwilym, Peter Hagoort, and Mark Dingemanse. 2016. “How iconicity helps people learn new words: neural correlates and individual differences in sound-symbolic bootstrapping.” Collabra 2 (1): 1–15. doi:10.1525/collabra.42.

Lupyan, Gary. 2008. “From chair to ‘chair’: A representational shift account of object labeling effects on memory.” Journal of Experimental Psychology: General 137 (2): 348–369. doi:10.1037/0096-3445.137.2.348.

Lupyan, Gary, and Daniel Casasanto. 2014. “Meaningless words promote meaningful categorization.” Language and Cognition FirstView: 1–27. doi:10.1017/langcog.2014.21.

Lupyan, Gary, David H. Rakison, and James L. McClelland. 2007. “Language is not Just for Talking: Redundant Labels Facilitate Learning of Novel Categories.” Psychological Science 18 (12): 1077–1083. doi:10.1111/j.1467-9280.2007.02028.x.

Lupyan, Gary, and Sharon L. Thompson-Schill. 2012. “The evocative power of words: Activation of concepts by verbal and nonverbal means.” Journal of Experimental Psychology: General 141 (1): 170–186. doi:10.1037/a0024904.

Monaghan, Padraic, Karen Mattock, and Peter Walker. 2012. “The role of sound symbolism in language learning.” Journal of Experimental Psychology: Learning, Memory, and Cognition 38 (5): 1152–1164. doi:10.1037/a0027747.

Nygaard, Lynne C., Debora S. Herold & Laura L. Namy. 2009. The Semantics of Prosody: Acoustic and Perceptual Evidence of Prosodic Correlates to Word Meaning. Cognitive Science 33(1). 127–146. doi:10.1111/j.1551-6709.2008.01007.x.

·

Basic reporting

This is a very nice and original study, I really liked the easiness of the procedure and the elegance that results from this in expalining method and results. I think this study would really add to the literature about sound-symbolism, and with no doubt deserves publication in PeerJ. However, before accepting it, there are a number of minor and major issues that need to be addressed - even if I want you to be sure by now that not only the minor, but also the major will be really easy to be solved by you).

Experimental design

The present manuscript describe original primary research within the Scope of the journal; clearly define relevant and meaningful research questions; the investigation was conducted rigorously and to a high technical standard, even if some methods might be described with more information in order to be reproducible by other researchers.
Finally, this research has been conducted in conformity with the prevailing ethical standards in the field.

Validity of the findings

The data seems to be robust, statistically sound, and controlled, even if the missing reporting of the means per condition leave me a little worried...

Additional comments

Main comment
This is a very nice and original study, I really liked the easiness of the procedure and the elegance that results from this in expalining method and results. I think this study would really add to the literature about sound-symbolism, and with no doubt deserves publication in PeerJ. However, before accepting it, there are a number of minor and major issues that need to be addressed - even if I want you to be sure by now that not only the minor, but also the major will be really easy to be solved by you).

Majors
p. 5: “the majority of research has focused on the mapping between a single experimental stimulus and a single label […]” - it might be useful here to more explicitly clarify that both names and figures where traditionally presented in pairs, and then briefly report also method and results of Nielsen & Rendall (2011) and Flumini, Ranzini, Borghi (2014).
p. 7: “Alternatively, encoding with symbolism may cause the individual items to be represented with richer feature representations, meaning that their individual shapes were recognized more efficiently […]” - I would add that this is probably also a consequence of a facilitation in accessing stored knowledge about each item which was sound-symbolically gruonded.
p. 8: “Alternatively Visual stimuli were abstract pencil drawings of complex shapes with ambiguous animacy […] we term these stimuli Vinčasso illustrations” - this is a really creative and funny name, but I would say something more about the rationale behind your choices for the shapes (for example why ambogous animacy? why pictures based on a so ancient drawing style?)… please try to give deeper arguments, or at least explain to me why you decided for omitting any explanation…
p. 9: “While the creature remained onscreen, participants were asked to guess which of the two categories it belonged to […]” - they are not “guessing”, they had to confirm they understood the label for the presented item…
p. 9: “In the second stage of training, labels were omitted” - so I think the procedure slightly changed… please describe it in detail… in general, it would be nice to specify the counterbalance of the leys and such aspects that are indispensable when some other lab would like to repeat the experiment and control for replication of results, e.g., in other languages, with younger participants etc. etc.
p. 9: footnote 1 - I understand the rationale behind the removal of this part of the study, however I’d like to have the chance to take a look into such results… and, even if I underline again that I’m perfectly well with what you said to expalin your decision, I think that in any case it might be better to report it at least in an appendix…
p. 10: “It is worth noting that the two categories are defined solely by their visual characteristics, making the labels an arbitrary, irrelevant feature of the training environment” - nice to hear that, and I imagine why it is so important… but I’m not sure researchers in other fields, or students, or any kind of non-expert audience, would catch the point… please clarify it.
p. 12: “F(1,40) […]” - first of all, change all “F” in “F” and all “p” in “p”, but more important, why do you report all Fs “1,40” instead of “1, 40”, “2, 78” etc.? and really fundamental, where can I find the means for each of your conditions?

Minors & typos
Abstract: “The aim of the present paper was to experimentally […]” - please correct “was” in “is”, an aim it is something that does not change in time, I think…
Abstract: “in labels with different range-of-reference […]” - please add “a” between “with” and “different”.
Abstract: “These findings reveal, that the systematicity […]” - please cut the coma between “reveal” and “that”.
p. 4: the abstract is reported again, be sure to correct in both pages, please…
p. 4: “sound-symbolism” (in the keywords list) - please be consistent along the whole paper in using the form “sound-symbolism” vs. “sound symbolism”…
p. 5: “Imai & Kita 2014” - please add a coma between the authors’ names and the year, as APA style requests, all along the paper.
p. 7: “The aim of the present paper was to experimentally […]” - the same as before… please correct “was” in “is”, thank you.
p. 8: “Eighty-two participants, students from a local Science Center and second-year undergraduate students at the Department of Psychology, of the University […]” - weird formulation, please simplify in “Eighty-two students […] participated […]”, and missing information not only about the University they are enrolled in, but also about sex, age etc.; please report that information in detail in your revised draft…
p. 8: “All participants reported normal or corrected-to-normal vision” - I really hope they were also naive as to the purposes of your study!
p. 8: “(e.g. ‘volab’)” - add a coma after “e.g.” along the whole paper please…
p. 12: “F(1,40) […]” - first of all, change all “F” in “F” and all “p” in “p”, but more important, why do you report “1,40” instead of “1, 81”, “2, 160” etc.?
p. 12: “[…] by the having previously encountered each item with labels which were congruent with the item’s visual appearance, or inhibited by having encountered incongruent labels” - please modify the period, possibily in a more straightforward formulation!
p. 13: “[…] but suggests that in the first block, participants may have been relying […]” - there is a not needed coma in this period… in general, be more careful with punctuation along the whole paper, please…
p. 15: “This finding is in line with, Gasser (2004) demonstrated […]” - just to give another example of what I just noticed…

---

## Round 0.2 · Minor Revisions

I invite you to address the minor points raised by the reviewer. I will then likely handle the paper myself, once it is revised.

·

Basic reporting

This is a re-review so I am looking at the revisions in light of the points raised in the original review and the authors’ revisions and response to it.

One of the important issues was grounding in the literature. This is now considerably improved. Another open question was the availability of stims. The authors have made clear these will be made available, which is good.

I’m afraid I still think the cloud metaphor is distracting because it’s not good enough an example of the actual phenomenon to be featured in the title. I’ll leave it to the editor to decide.

Smaller matters of basic reporting included the figures, and it is good to see that most suggestions have been followed. The new figures are useful, though graphical indications of group means might be a useful addition. A small detail related to the figures: the study showing individual differences in sound-symbolic sensitivity is Lockwood et al. 2016 in Collabra (this is misattributed in footnote 4).

Experimental design

The authors have carried out a rating experiment to address some of my reservations about the visual stimuli.

Most questions about auditory stimuli have also been answered by the revisions. However, it remains unclear whether the speaker recording the stimulus items was aware of the experimental manipulation.

The case of the excluded condition is explained a bit more clearly in the author’s response, and quite nicely in the “curious case” document. I would recommend that a little more of the explanation be carried over into the ms, since as it is, many readers will be confronted with the same questions as the reviewers in the initial round.

Validity of the findings

The predictions and expected direction of effects have been made more clear, although I think it would be useful to make a clear distinction between prior hypotheses and post-hoc rationalizations.

Some of the challenges with interpreting the findings remain, due to the equivocal use of the terms ‘systematicity’ and ‘sound symbolism’.

With regard to what sound symbolism is, the ms still does not seem to make up its mind. Is it just a form of systematicity, i.e. merely a statistical regularity which may as well be language-specific? Or is it a kind of iconicity, i.e., grounded in perceptual analogies and cross-modal associations? Most parts of the ms seems to push for the second view: sound symbolism is framed as a generalized form of crossmodal congruence (p. 5), the lit review features work that sees sound symbolism as a form of iconicity (e.g. Imai, Kita, Nygaard, Berlin), and the very last sentence of the paper allows that there are “non-sound-symbolic but otherwise systematic labels”, implying a distinction between sound-symbolism and systematicity (p. 18). But confusingly, a full page of the Discussion (p. 17) seems to propose that sound symbolism is just a form of ‘systematicity’, watering down the notion to any kind of regularity in form-meaning mapping.

This watered down notion is in contrast with most recent work, which makes a distinction between systematicity (statistical regularities between form and meaning, often language-specific) and iconicity (resemblance between form and meaning, often shared across languages because grounded in crossmodal congruence). Recent findings show that (i) iconicity accounts better for age of acquisition effects formerly attributed to systematicity (Massaro and Perlman 2017), and (ii) systematicity and iconity are not correlated in large lexical corpora (Winter et al. 2017).

Judging from the proposal that the findings are rooted in crossmodal congruence, it seems the authors don’t have a well-defined notion of systematicity in mind (or certainly not one that aligns with the recent literature). It seems the findings are best understood as involving iconic affordances rather than mere statistical regularities. To be precise, the findings appear to reveal that certain speech sounds have iconic or sound-symbolic affordances that may render them more or less congruent for the expression of certain categorial or referential meanings. The authors do themselves a disservice by chalking it up to a vague notion of ‘systematicity’ which recent work has shown to be language-specific and distinct from cross-modally grounded iconicity.

Refs cited:
Massaro, Dominic W., and Marcus Perlman. 2017. “Quantifying Iconicity’s Contribution during Language Acquisition: Implications for Vocabulary Learning.” Frontiers in Communication 2. doi:10.3389/fcomm.2017.00004.
Winter, Bodo, Marcus Perlman, Lynn Perry, and Gary Lupyan. 2017. “Which words are most iconic? Iconicity in English sensory words.” Interaction Studies.

---

## Round 0.3 · accepted · Accept

Your article has been accepted for publication of PeerJ. Congratulations!